# Peeking into the Stingers: A Comprehensive SWATH-MS Study of the European Hornet *Vespa crabro* (Linnaeus, 1758) (Hymenoptera: Vespidae) Venom Sac Extracts

**DOI:** 10.3390/ijms25073798

**Published:** 2024-03-28

**Authors:** Xesús Feás, Manuela Alonso-Sampedro, Susana Belén Bravo, Carmen Vidal

**Affiliations:** 1Academy of Veterinary Sciences of Galicia, 15707 Santiago de Compostela, Spain; xesusfeas@gmail.com; 2Fundación Instituto de Investigación Sanitaria de Santiago de Compostela (FIDIS), Hospital Clínico, 15706 Santiago de Compostela, Spain; manuela.alonso.sampedro@sergas.es (M.A.-S.); sbbravo@gmail.com (S.B.B.); 3Research Methods Group (RESMET), Health Research Institute of Santiago de Compostela (IDIS), University Hospital of Santiago de Compostela, 15706 Santiago de Compostela, Spain; 4Network for Research on Chronicity, Primary Care, and Health Promotion (RICAPPS-ISCIII/RD21/0016/0022), University Hospital of Santiago de Compostela, 15706 Santiago de Compostela, Spain; 5Proteomic Unit, Health Research Institute of Santiago de Compostela (IDIS), University Hospital of Santiago de Compostela, 15706 Santiago de Compostela, Spain; 6Allergy Department, University Hospital of Santiago de Compostela, 15706 Santiago de Compostela, Spain; 7Department of Psychiatry, Radiology, Public Health, Nursing and Medicine, Faculty of Medicine, University of Santiago de Compostela (USC), 15782 Santiago de Compostela, Spain

**Keywords:** *Vespa crabro*, European hornet, wasps, venom sac, castes, gynes, workers, proteomics, SWATH-MS analysis, venomics

## Abstract

This study aimed to investigate the venom sac extracts (VSEs) of the European hornet (EH) *Vespa crabro* (Linnaeus, 1758) (Hymenoptera: Vespidae), focusing on the differences between stinging females, gynes (G), and workers (W), at the protein level. Using a quantitative “Sequential Window Acquisition of all Theoretical Fragment Ion Mass Spectra” (SWATH-MS) analysis, we identified and quantified a total of 240 proteins. Notably, within the group, 45.8% (*n* = 110) showed significant differential expression between VSE-G and VSE-W. In this set, 57.3% (*n* = 63) were upregulated and 42.7% (*n* = 47) downregulated in the G. Additionally, the two-hundred quantified proteins from the class Insecta belong to sixteen different species, six of them to the Hymenoptera/Apidae lineage, comprising seven proteins with known potential allergenicity. Thus, phospholipase A1 (Vesp v 1), phospholipase A1 verutoxin 2b (VT-2b), hyaluronidase A (Vesp v 2A), hyaluronidase B (Vesp v 2B), and venom allergen 5 (Vesp v 5) were significantly downregulated in the G, and vitellogenin (Vesp v 6) was upregulated. Overall, 46% of the VSE proteins showed differential expression, with a majority being upregulated in G. Data are available via ProteomeXchange with identifier PXD047955. These findings shed light on the proteomic differences in VSE between EH castes, potentially contributing to our understanding of their behavior and offering insights for allergy research.

## 1. Introduction

A wide array of venomous creatures can be found across various phyla, showcasing an extraordinary variety of taxa, toxins, targets, clinical effects, and outcomes [1]. In fact, venomous organisms have captivated the scientific community due to their remarkable chemical adaptations and diverse biological functions [2,3]. Proteomics, the large-scale study of proteins within a biological sample, has emerged as a powerful tool to decipher the intricate molecular components of venoms. By combining advanced analytical techniques, such as liquid chromatography–mass spectrometry (LC-MS), with bioinformatics approaches, researchers can obtain detailed insights into the venom’s proteinaceous constituents [4,5,6]. This in-depth understanding of the venom proteome allows the identification, characterization, and functional annotation of venom proteins, enabling a holistic understanding of the venom’s biological activities and potential applications in various fields [7].

*Vespa crabro* (Linnaeus, 1758) (Hymenoptera: Vespidae), commonly known as the European hornet (EH), is one of the largest species of hornet, measuring up to 35 mm in length, and is found in Europe, Asia, and North America [8]. EHs are social insects that live in large colonies. They build their nests in tree hollows, cavities in buildings, or other protected areas. The nest is made of a paper-like material created with wood fibers which are chewed up and mixed with saliva [9]. As predators, EHs feed their brood with a variety of insects, including other bees and wasps, flies, and caterpillars. Adults are also attracted to sweet substances and may be seen feeding on ripe fruit or sugary liquids [10]. While EHs can be beneficial for controlling other insect populations, they can also be a nuisance or a danger to humans and other animals [11]. Their sting is painful, and some people sensitized against their venom may develop an allergic reaction after suffering from a sting [12,13,14,15,16,17,18,19,20].

Research has been conducted to identify and characterize the bioactive components present in the venom of wasps in general and EH in particular. In general, wasp venoms are composed of a diverse array of peptides, proteins, and enzymes that play crucial roles in prey capture, defense, and communication within the colony [21,22]. Studies have shown that EH venom components like crabrolin and mastoparans exhibit antimicrobial properties against various pathogens, including bacteria and fungi [23,24,25,26]. Furthermore, the venom of EH contains enzymes, such as hyaluronidase, which play a role in facilitating the spread of venom within the prey or victim’s tissues [27]. Moreover, the immunological properties of EH venom compounds have been investigated. Allergenic components, such as antigen 5 molecules, have been identified and characterized, shedding light on the structural basis of antigenic cross-reactivity among different hymenopteran venoms [28,29,30]. By investigating the venom compounds of different wasps, researchers aim to gain insights into the molecular mechanisms underlying their venomous properties and explore potential applications in various fields, including pharmacology and biotechnology [31,32,33,34].

Currently, research on the venom composition of social insects, particularly wasps, has gained attention due to the multifaceted functions of venom beyond prey capture and defense. However, these studies have been limited by their inability to quantify the complete abundance of venom components accurately. Moreover, investigations into the caste-specific differences in venom composition have been scarce [35], despite the behavioral and physiological variations observed among female castes in wasp colonies.

To address these knowledge gaps and build upon the existing body of research, our study applies a quantitative Sequential Window Acquisition of all Theoretical Fragment Ion Mass Spectra (SWATH-MS) approach to comprehensively analyze and compare the venom sac extract (VSE) proteome of gyne (G) and worker (W) EH females. SWATH-MS is a mass spectrometry-based technique that allows for comprehensive and quantitative proteomic analysis [36,37]. It is a data-independent acquisition method that offers advantages over traditional mass spectrometry quantitative methods, such as selected reaction monitoring or parallel reaction monitoring. In the present work, by quantifying the abundance of proteins from venom sac extracts (VSEs) of gynes (VSE-G) and workers (VSE-W), we aim to uncover potential caste-specific differences within the EH social structure.

## 2. Results

### 2.1. Venom Sac, Extract Protein Quantitation, and SDS-PAGE

The venom sac (VS) was carefully removed from both gyne (designated as VS-G) and worker (designated as VS-W) stinging apparatuses. After being cleaned, a distinct contrast in the external structure of the castes’ VS was observed (as shown in Figure 1). Specifically, the VS-W appeared elongated and exhibited greater transparency compared to the more rounded and opaque appearance of the VS-G.

The protein of the venom sac extracts (VSEs) of gynes (VSE-G) and workers (VSE-W) was quantified (Figure 2a), and no differences (Mann–Whitney test *p* > 0.99) were found between G (186.9 ± 29 μg/sac) and W (188.8 ± 89 μg/sac). Figure 2b illustrates the electrophoretic separation of VSE-G and VSE-W. The analysis unveiled a diverse composition of venom components, spanning molecular masses from 10 to 250 kDa. Notably, the VSE-G profile displayed a higher concentration of proteins within the 25–250 kDa mass range compared to the VSE-W, while no variations were detected within the low-molecular-mass range (10–25 kDa). Evident and distinct bands corresponding to known potential allergens of EH (Vesp c 2, two isoforms of Vesp c 2 and Vesp c 5) were pinpointed based on their molecular masses in VSE-W, with these bands being less pronounced in VSE-G.

### 2.2. Proteomic Quantitative Analysis of Total Proteins in the Venom Sac Extract

To delve deeper into the distinctions within VSE content between gynes (G) and workers (W), we conducted LC-MS/MS mass spectrometry, utilizing the SWATH-MS quantification approach to quantified proteins with varying expression. Across the four biological replicates, a cumulative count of 240 proteins were both identified and quantified (detailed in Table 1), encompassing 11 distinct classes and 33 different species. As anticipated, a significant majority of the identified proteins, constituting 83.3% (*n* = 200), were categorized under the class Insecta.

Out of the 240 proteins identified, a substantial proportion, accounting for 45.8% (*n* = 110), exhibited notable differences in expression between the VSE-G and VSE-W samples (with fold change (FC) ≥ 1.5 and *p* ≤ 0.05). Specifically, within this subset, 57.3% (*n* = 63) demonstrated upregulation, while 42.7% (*n* = 47) displayed downregulation in the VSE-G samples compared to VSE-W. To visually depict the overall quantification of these proteins across female castes and highlight the dysregulated ones, a volcano plot (refer to Figure 3) was utilized. Detailed information regarding the SWATH-MS most-likely ratio (MLR)-normalized areas (per protein and sample), as well as fold change and *p*-values (determined by *t*-test), can be found in Appendix A.

Among the sixty-three proteins upregulated in the VSE-G, the four most overexpressed are Glycogen debranching enzyme (FC = 40.1; *p* = 0.004), muscle LIM protein MIp84B isoform X2 (FC = 34.6; *p* = 0.016), myosin light chain alkali (FC = 30.3; *p* = 0.039), and vitellogenin (FC = 17.8; *p* = 0.0005). Regarding the four most downregutaled proteins in the VSE-G, we observed a decreased expression of venom allergen 5 (FC = 38.2; *p* = 0.006), phospholipase A1 (FC = 19.2; *p* = 0.0003), calmodulin (FC = 19.2; *p* = 0.025), and alpha-glucosidase (FC = 15.0; *p* < 0.0001).

Figure 4 shows the most common gene ontology (GO) terms of the differentially expressed proteins in EH female castes, allowing for an assessment of the biological processes (Figure 4A) in which they are involved, and a quick comparison of protein functions at the molecular level (Figure 4B). The results revealed the involvement of the VSE-G downregulated proteins in the regulation of carbohydrate metabolic process, lipid metabolic process, defense response, glucose metabolic process, and proteolysis, as well as the participation of the VSE-W overexpressed proteins in the glycogen metabolic process, protein phosphorylation, tricarboxylic acid cycle, and cell differentiation. Regarding the molecular function, the results revealed the involvement of the VSE-G upregulated proteins in ATP binding, metal ion binding, calcium ion binding, magnesium ion binding, and pyridoxal phosphate binding. On the other hand, the VSE-W upregulated proteins are involved in NAD binding, lipid binding, hyalurononglucasaminidase activity, and phospholipase A1 activity. A detailed list with GO terms is available in Appendix A.

### 2.3. Proteomic Quantitative Analysis from Dysregulated Proteins of the Class Insecta

The two-hundred quantified proteins from the class Insecta belong to sixteen different species (Table 1), six of them to the Hymenoptera/Apidae lineage. Among them, we were able to quantify seven proteins with known potential as allergens (shown as blue spots in Figure 3). As shown in Figure 5, phospholipase A1 (Vesp v 1), phospholipase A1 verutoxin 2b (VT-2b), hyaluronidase A (Vesp v 2A), hyaluronidase B (Vesp v 2B), and venom allergen 5 (Vesp v 5) were significantly downregulated in the VSE-G, and vitellogenin (Ves v 6) was upregulated (Mann–Whitney test, *p* < 0.05).

In general, 46% (*n* = 92) of the proteins from class Insecta showed significant differential expression between VSE-G and VSE-W samples (FC ≥ 1.5 and *p* ≤ 0.05); 65% (*n* = 60) were upregulated and 35% (*n* = 32) downregulated in the VSE-G (Appendix A, respectively). From the Hymenoptera/Apidae lineage, a total of 44, 43, 36, 29, and 28 proteins from *Bombus terrestris*, *Apis mellifera*, *Apis cerana*, *Dufourea novaeangliae,* and *Frieseomelitta varia*, respectively, were quantified in the VSE of EH female castes.

From the *D. novaeangliae* (family Halictidae), we quantified 29 proteins, showing 48.3% (*n* = 14) differential expression between the EH female castes. Eleven were upregulated in the VSE-G and three were downregulated in the VSE-G. All the proteins quantified from *D. novaeangliae* are available in STRING, the protein–protein interaction network tool (accessed on 16 June 2023). In this way, a map of the interaction of proteins and their main biological function was obtained (Figure 6). We performed k-means clustering analysis and obtained three clusters, with eight, ten, and eleven proteins. The three downregulated proteins in VSE-G all belong to cluster 3 (Figure 6A,D), with the upregulated proteins mainly being in clusters 1 (six proteins) and 2 (four proteins). We also carried out a search for biological processes using the functional enrichment analysis (FunRich) tool (version 3.1.4). The results revealed the involvement of the upregulated proteins in the generation of precursor metabolites and energy and the actomyosin structure organization, while the downregulated proteins are involved in the glicerol-3-phosphate metabolic process.

### 2.4. Proteomic Quantitative Analysis from Dysregulated Proteins of the Other Classes

A total of 40 proteins from the other classes were quantified in the VSE-G and VSE-W. Among them, 45% (*n* = 18) showed significant differential expression between VSE-G and VSE-W samples (FC ≥ 1.5 and *p* ≤ 0.05); 16.7% (*n* = 3) were upregulated and 83.3% (*n* = 15) downregulated in the VSE-G (Table 2). A total of 17 (42.5%) proteins belonged to the mimic poison frog *R. imitator*; among them, 47% (*n* = 8) were downregulated in the VSE-G.

## 3. Discussion

The selective sampling approach of collecting specimens exclusively from a single colony offers a unique advantage, ensuring a cohesive genetic and environmental context for the study. Moreover, the integration of cutting-edge analytical techniques, by the application of SWATH-MS, with its capacity for quantitative analysis of the complete proteomic profile, allows us to explore the differences in venom composition across castes with unprecedented precision. We showed, for the first time, differences in the external morphology and in the protein-binding pattern between venom sacs (VSs) and venom sac extracts (VSEs) from female castes of the EH. These differences were similar to those found in *Vespa velutina* [35]; thus, these two hornet species exhibit a high degree of conservation of the venom sac between female castes. Previous work has shown similar results for *Vespa orientalis* (an egg-shaped venom sac, 3–5 mm length and 2–3 mm at maximum width) [38]. In contrast, other authors indicated venom sacs of *V. velutina* to be approximately 1 mm in length, white, and transparent [39]. The electrophoretic separation of VSE highlights differences in their profiles, suggesting functional specialization and adaptation to their distinct roles within the colony. Gynes appear to possess a higher diversity or abundance of proteins in the 25–250 kDa mass range, potentially related to reproductive and colony establishment functions. In contrast, workers display clear and distinct bands of known allergenic proteins, indicating a focus on defense and deterring potential threats. These findings underscore the caste-specific roles and ecological interactions of the EH, shedding light on the complex nature of their venom composition. Further research is needed in more hornet species to identify and characterize specific venom components, enhancing our understanding of the functional significance and mechanisms behind these differences.

Since the EH protein database is not well-annotated, protein and peptide identification was based on homology of sequence, and was carried out using a customized database including *Vespa* + *Vespa velutina* + *Apis mellifera* + poison + toxins (available online: https://www.uniprot.org/ (accessed on 2 February 2023)). A total of 240 proteins were quantified in the VSE-G and VSE-W, belonging to 11 different classes, 14 orders, 17 families, and 33 species, highlighting *Insecta* (*n* = 200); *Amphibia* (*n* = 17); *γ-Proteobacteria* (*n* = 8); and *Arachnida* (*n* = 5).

Proteins belonging to class *Insecta* were quantified by SWATH-MS in the VSE-G and VSE-W. *F. varia* (*n* = 28 proteins), also known as the yellow marmalade bee, is a highly eusocial aggressive stingless bee distributed in several areas of Brazil. Workers are completely sterile, with a programmed cell death during pupal development, resulting in non-functional ovaries. *A. cerana* (*n* = 36 proteins), the Asian honey bee, is a species native to South, Southeast, and East Asia, territory where 22 species of hornets are present. Hornets and their native honey bee prey are the main characters of a coevolutionary arms race that is made evident by the conspicuous number of reciprocal adaptations evolved by both animals. *D. novaeangliae* (*n* = 29 proteins), the sweat bee, is a soil-dwelling solitary species found in the northeastern USA and Canada, and forages for pollen only on pickerel weed (*P*. *cordata*). Prior research indicates the possibility that the entire existing Hymenoptera lineage could have originated from a shared ancestor with venomous traits [40].

Proteins were quantified by SWATH-MS from *R*. *imitator* (*n* = 17), the mimic poison frog, naturally distributed in the north-central region of eastern Peru. *R. imitator* mimics not one but three other *Ranitomeya* species of highly toxic poison frogs (*R. fantastica*, *R. variabilis*, and *R. ventrimaculata*). This represents the only known example of mimetic radiation in amphibians. Also, this species shows pair bonding and biparental care, being the only known monogamous amphibian in the wild confirmed by paternity analysis [41]. Additional investigation is required to determine if these proteins pose a potential threat to mammals following EH stings.

Proteins from *γ-proteobacteria* (*Klebsiella* sp. *RIT-PI-d*, *Cronobacter sakazakii*, *Citrobacter freundii*, *Escherichia coli*) and *Bacilli* (*Lactobacillus apis*, *Lactococcus lactis*) were identified by SWATH-MS in the VSE of EH. The gut bacteria found in various species and regions within the Vespa genus exhibited significant similarity in their compositions at both the phylum and class levels [42]. In fact, *Lactobacillus* sp. [43], which are the main species in the gut of honey bees, are also present in EH. *Klebsiella* sp. [44] was also previously reported in *Vespa* sp. individuals. Further investigation is crucial to determine the source of these microorganisms, some of which are commonly found in hornet gut flora or could potentially contaminate the analyzed extracts during the sac extraction process. Two proteins originating from *Tropilaelaps mercedesae* (Arachnida) were quantified. *Tropilaelaps* mites typically parasitize giant honey bees such as A. *dorsata*, *A. laboriosa*, and *A. breviligula*. Among the four species of *Tropilaelaps*, *T. clareae* and *T. mercedesae* have successfully adapted to *A*. *mellifera*. *T. mercedesae* has recently expanded its geographic distribution to regions in South Korea and China [45].

The invasive species *V*. *velutina*, commonly known as the Asian hornet, has garnered significant attention both in the media and the medical community due to its expanding presence and the potential health implications associated with its venom [46,47,48]. Media coverage has often emphasized the negative impact of *V*. *velutina* on honey bee populations and its potentially aggressive behavior towards humans, amplifying public awareness. From an allergological point of view, a comparison between *V*. *velutina* [35] and EH highlights the difference at the level of phospholipase A1 (Vesp v 1), where two distinct bands are identified [designated as Vesp v 1 (a) and (b)]. It is notable that in *V*. *velutina,* the Vesp v 1 (a) band is the most predominant, while in EH, the most abundant band is Vesp c 1 (b). In addition, VSE-G have been observed to have higher numbers of Vesp c 2 and Vesp c 1 compared to *V. velutina* gynes.

While this study has provided valuable insights into the proteomic differences within the VSE of *V*. *crabro* castes, there are certain limitations that warrant consideration. One limitation lies in the fact that the analysis focused on homogenized venom sac extracts, potentially overlooking variations that may arise from different extraction methods [49]. To fully comprehend the allergological implications of these distinctions, future research should employ IgE immunoblot analyses to evaluate the specific immunoreactivity of these proteins and their potential contribution to allergic responses [50,51,52]. An optimal approach for advancing our understanding of hymenopteran envenomation and its implications would involve comprehensive proteomic analyses encompassing various hymenopteran species within a specific geographical region. This holistic approach aligns with the broader One Health perspective, integrating entomological, medical, and ecological dimensions to address the multifaceted challenges posed by stinging insects.

## 4. Materials and Methods

### 4.1. Source of Insects

#### 4.1.1. Study Area

Field research aimed at collecting the *V*. *crabro* insects was conducted in the Galician municipality of Pazos de Borbén. Galicia is located in the northwest of Europe on the Iberian Peninsula. The municipal term of Pazos de Borbén has an extension of 50 km^2^ and is located between 42°14′40″ north latitude and 4°47′30″ and 4°54′40″ west latitude (Appendix A). The climate is typified as humid oceanic with a tendency for summer aridity (average annual temperature of 13 °C, and an average annual precipitation of 1994 mm). Rock substrate is composed of granite of two mica. The most abundant soils are the brown earths and the rankers. Considering the flora, the native forest is mainly composed of *Quercus robur*, and accompanied by *Quercus pyrenaica*, *Castanea sativa*, *Laurus nobilis*, and *Fagus sylvatica,* with a herbaceous layer formed by *Teucrium scorodonia*, *Viola riviniana*. *Luzula forsteri,* or *Luzula sylvatica*. Riparian forest, bordering the river courses (Borbén, Asemonde, Pozo Negro, Pequeno, and Alvedosa), is dominated by *Alnus glutinosa*, and accompanied by *Betula pubescens* subsp. *celtibérica*, various species of *Salix* sp., and even *Q. robur*. Large patches of Ulex europaeus, *Rubus* sp., *Pteridium aquilinum,* and *Cytisus striatus* are also found. Ascending in height, other species appear, such as *Genista florida*. *Cytisus scoparius,* or *Cytisus grandiflorus*. Reforestation productive forest is formed by E. *globulus*, P. *pinaster,* and mixed masses of the two species.

#### 4.1.2. Insect Localization, Identification, and Nest Colony Composition

*Eucalyptus* logging points were identified within the study area and strategically selected as potential hotspots for vespid activity (Appendix A). The identification of *V*. *crabro* individuals was conducted based on their external morphological characteristics. They are easily distinguished from V. *velutina* by coloration (Appendix A). Upon successful identification of V. *crabro* individuals at the eucalyptus cutting points, tracking was initiated. Individual hornets were followed to trace their flight paths and possible nesting locations. The nest of the *V*. *crabro* was collected on 5 December 2022 during the night (Appendix A). The count of mature individuals was conducted manually to determine insect adult population (IAP). Subsequently, the collected insects were sorted based on the categories of sex, reproductive individuals, and castes: gyne (G), male (M), queen (Q), and worker (W). Female and male hornets exhibit sexual dimorphism. Female hornets have 12 antennal segments (including the scape and pedicel), while male hornets have relatively long, curly-ended antennae with 13 segments. Female hornets have 6 gastral segments and are equipped with a stinger (modified ovipositor used for defense). On the other hand, male hornets have 7 exposed gastral segments and lack a stinger. Males have a bilobate apex of the last sternite, which appears sharp in females. The distinction between workers and gynes (future queens) was established based on the size, the gynes being larger than the workers (Figure 7). The sizes were as follows: workers, 15–25 mm; queens, 25–35 mm. The insect adult population (IAP) found (gyne, male, queen, and worker) was as follows: *n* = 168 individuals (58, 42, 1, and 67). A detailed description of the sampling area, insect localization and identification and nest removal can be found in Appendix A.

### 4.2. Venom Sac Extract Obtention

The venom sacs (VSs) were collected from frozen insects by carefully extracting the stinger apparatus from the hornet’s abdomen using forceps. Subsequently, the VSs were separated from the stingers. Three VSs were combined to create four pools of worker (VS-W) and gyne (VS-G), which were then homogenized in phosphate-buffered saline (PBS). The sample size was as follows: gynes = 12 (1–4 pooled samples; each sample contains venom sacs from three individuals); workers = 12 (1–4 pooled samples; each sample contains venom sacs from three individuals). Any remaining tissues were eliminated by centrifugation at 3000 rpm for 3 min. The resulting venom sac extract from workers (VSE-W) and gynes (VSE-G) was transferred to new tubes and stored at −20 °C, until further analysis.

### 4.3. Venom Sac Extract Protein Quantitation

The protein content of the VSE-W and VSE-G was quantified using the Bradford method from Bio-Rad, Hercules, CA, USA [53].

### 4.4. Sodium Dodecyl Sulfate Polyacrylamide Gel Electrophoresis (SDS-PAGE)

Analysis of the VSE proteins was conducted through 12% SDS–PAGE under reducing conditions, followed by staining with Sypro Ruby (Bio-Rad, Hercules, CA, USA). Finally, the protein bands were visualized using the ChemiDoc MP imaging system (Bio-Rad, Hercules, CA, USA).

### 4.5. Sample Preparation for Mass Spectrometric Analysis and Quantification Using Sequential Window Acquisition of All Theoretical Mass Spectra (SWATH-MS)

A total of 24 μg of protein was concentrated within a single band on an SDS–PAGE gel and subjected to manual digestion. Subsequently, the resulting peptides were dissolved in 0.1% formic acid for further analysis following established procedures [35]. For quantitative proteomic analysis, we employed a hybrid quadrupole-TOF mass spectrometer 6600+ (SCIEX, Framingham, MA, USA) coupled with a micro-liquid chromatography (LC) system Ekspert nLC425 (Eksigen, Dublin, CA, USA). Data acquisition utilized ProteinPilot v.5.0.1, PeakView v.2.2, MarkerView, and SWATH Acquisition MicroApp v.2.0 software (SCIEX, Framingham, MA, USA). A customized database encompassing *Vespa*, *Vespa velutina*, *Apis mellifera*, venom, and toxins Uniprot databases [available online: https://www.uniprot.org/ (accessed on 2 February 2023)] was utilized. Peptide mixtures from sample pools were chromatographed for 40 min using data-dependent acquisition in positive ion mode to generate MS/MS spectral libraries. A false discovery rate of 1%, with confidence scores exceeding 99%, was set. This spectral library was then employed to establish the spectral window acquisition for SWATH-MS analysis. Subsequently, 4 μL from each sample was individually analyzed. The SWATH-MS method involved data acquisition in a cycle of 100-flight mass spectrometry (TOF MS/M) windows, with quantitative analyses supported by extracted ion chromatograms at both MS1 and MS2 levels. Proteins with more than 10 peptides and seven transitions were selected for quantification, excluding shared or modified peptides. To compare *Vespa crabro* caste groups (VSE-G and VSE-W), Student’s *t*-test analysis was conducted based on the averaged area sums of all transitions derived for each protein. The resulting *p*-values were used to determine the significance of differences between the two groups. Proteins exhibiting a *p*-value less than 0.05, with a 1.5-fold increase or 0.66-fold decrease, were considered differentially expressed.

### 4.6. Protein Functional Analysis

The quantified proteins underwent comprehensive functional analysis to elucidate their involvement in molecular functions, biological processes, cellular components, and protein families. These proteins were systematically interrogated against the UniProtKB protein databases [available online: http://www.uniprot.org (accessed on 10 February 2023)]. Subsequently, the UniProt codes corresponding to proteins from D. novaeangliae were meticulously analyzed utilizing the STRING: Functional protein association network [available online: http://www.string-db.org (accessed on 14 June 2023)]. This approach facilitated the construction of interaction maps, execution of cluster analysis, and exploration of underlying biological processes.

### 4.7. Statistical Analysis

Graphical representations comparing the SWATH-normalized area of expressed proteins in the VSE-G and VSE-W were created using box plots, presenting the median and whiskers (spanning from minimum to maximum values), with black dots indicating individual sample values. Furthermore, a Volcano plot was constructed by juxtaposing the log2 fold change of the identified proteins against their respective adjusted −log10 *p*-values. Protein up- or downregulation was determined based on a fold change (FC) ≥ 1.5, with statistical significance attributed to a *p*-value of ≤0.05 across all tests. Graphical analyses were conducted utilizing GraphPad Prism software (version 9.5.0) (GraphPad Software, San Diego, CA, USA).

## Figures and Tables

**Figure 1 ijms-25-03798-f001:**
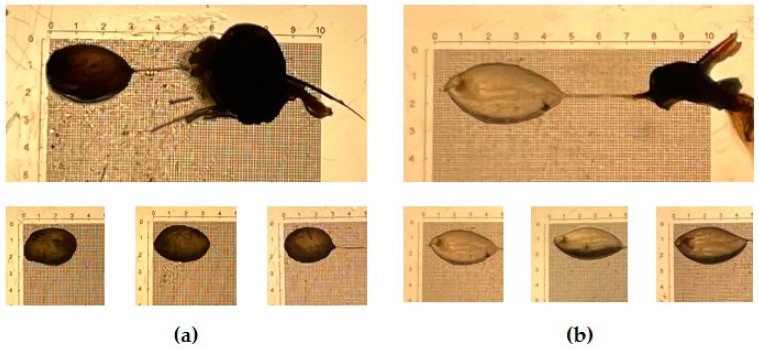
*Vespa crabro* stinging apparatus (up) and venom sac (down) of (**a**) gynes and (**b**) workers. Scale in mm. Photo Author: M. Alonso-Sampedro.

**Figure 2 ijms-25-03798-f002:**
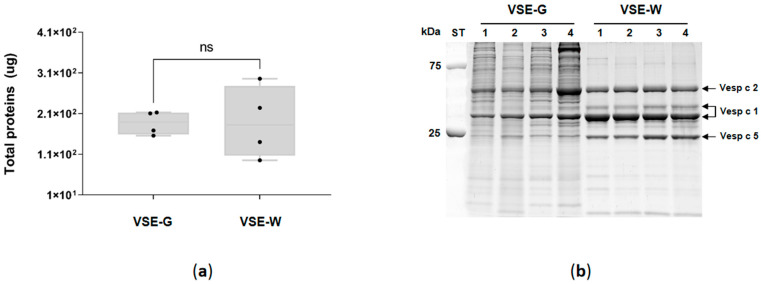
Analysis of the venom sac extracts (VSEs) of gynes (VSE-G) and workers (VSE-W) of *Vespa crabro*; (**a**) box plot with median, whiskers min to max, and black dots for the individual values of total proteins in micrograms (μg) per venom sac (ns, no statistical significance for Mann–Whitney test); (**b**) SDS-PAGE arrows indicate the bands corresponding to the known potential allergens according to their molecular mass: Vesp c 5, venom allergen 5 (23 kDa); Vesp c 1, phospholipase A1 (34 kDa); and Vesp c 2, hyaluronidase (40 kDa). kDa, kilodalton; ST, precision plus protein standard.

**Figure 3 ijms-25-03798-f003:**
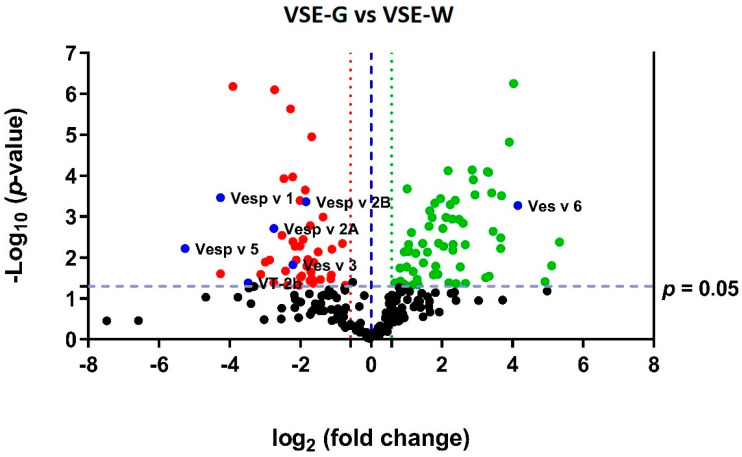
Volcano plot of the venom sac extract (VSE) quantitative proteomics data of the *Vespa crabro* female castes (gynes and workers). VSE proteins (shown as spots) are ranked according to their statistical *p*-value (*y*-axis) as −log_10_ and their relative abundance ratio as log_2_ fold change (*x*-axis). Spots positioned off-center indicate the most significant variations between the venom sacs of gynes (G) and workers (W). Significant changes are determined by cut-offs: fold change ≥ 1.5 (indicated by dashed lines in red and green) and *p* ≤ 0.05 (determined by *t*-test). In the representation, green spots highlight upregulated proteins in VSE-G, red spots indicate downregulated proteins in VSE-G, and black spots represent proteins that show no dysregulation between the two groups. Noteworthy potential allergens are marked as blue spots, including Vesp v 5 (venom allergen 5), Vesp v 1 (phospholipase A1), Vesp v 2A (hyaluronidase A), Vesp v 2B (hyaluronidase B), VT-2b (phospholipase A1 verotoxin-2b), Ves v 6 (dipeptidylpeptidase IV), and Ves v 6 (vitellogenin). VSE-G refers to venom sac extracts of gynes, while VSE-W refers to venom sac extracts of workers. Protein nomenclature is based on software identification.

**Figure 4 ijms-25-03798-f004:**
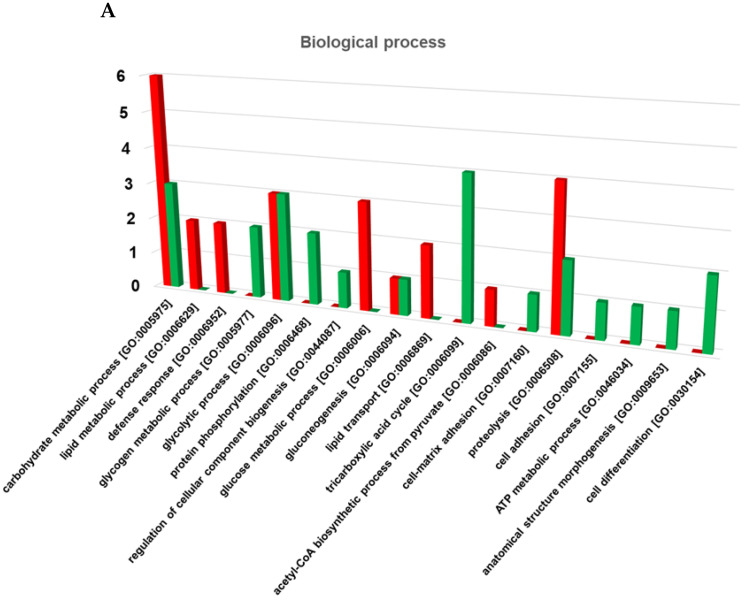
(**A**) Biological processes and (**B**) molecular functions mainly related to the differentially expressed proteins in the venom sac extracts (VSEs) of *Vespa crabro* gynes (G) and workers (W). The histograms depict the primary categories corresponding to each gene ontology (GO) term wherein differentially expressed proteins participated (*p* < 0.05). The *y*-axis illustrates the count of individual proteins within each GO term, while the *x*-axis denotes the GO term itself. Red signifies VSE-G downregulated proteins, while green indicates VSE-G upregulated proteins. The nomenclature of proteins is based on software identification.

**Figure 5 ijms-25-03798-f005:**
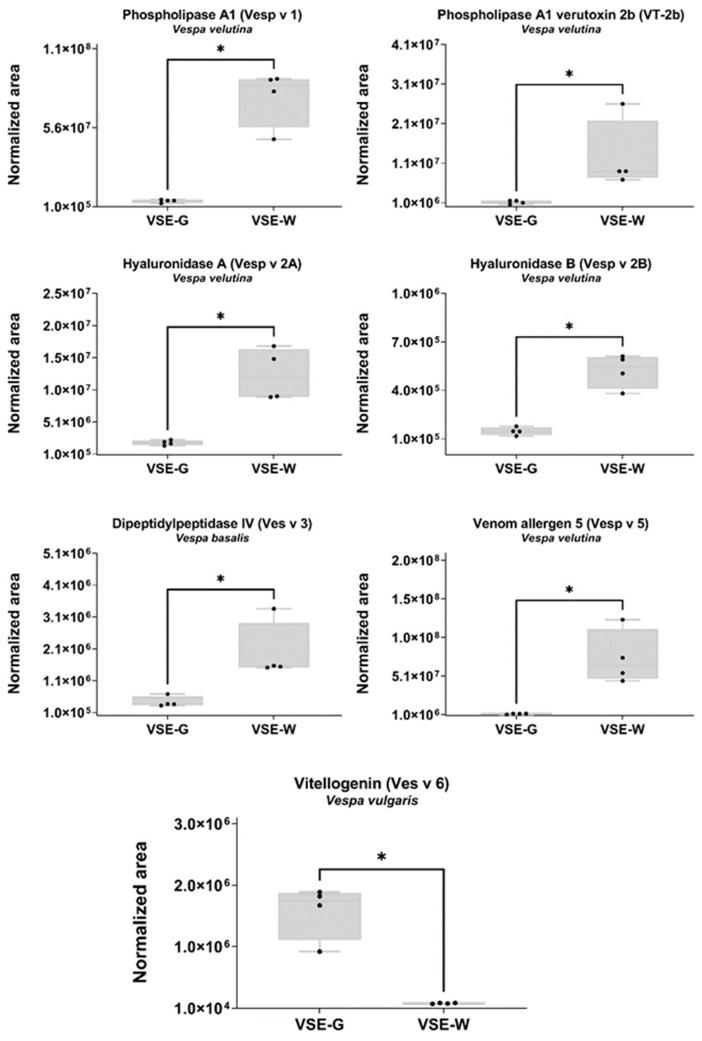
Representation of the SWATH-MS normalized area of the known potential allergens in the venom sac extract (VSE) of the *Vespa crabro* gynes (G) and workers (W). Box plot with median, whiskers min to max, and black dots for the individual values of the normalized area. Statistical differences by Mann–Whitney test, * *p* < 0.05. VSE-G: venom sac extracts of gynes; VSE-W: venom sac extracts of workers. The nomenclature of proteins is based on software identification.

**Figure 6 ijms-25-03798-f006:**
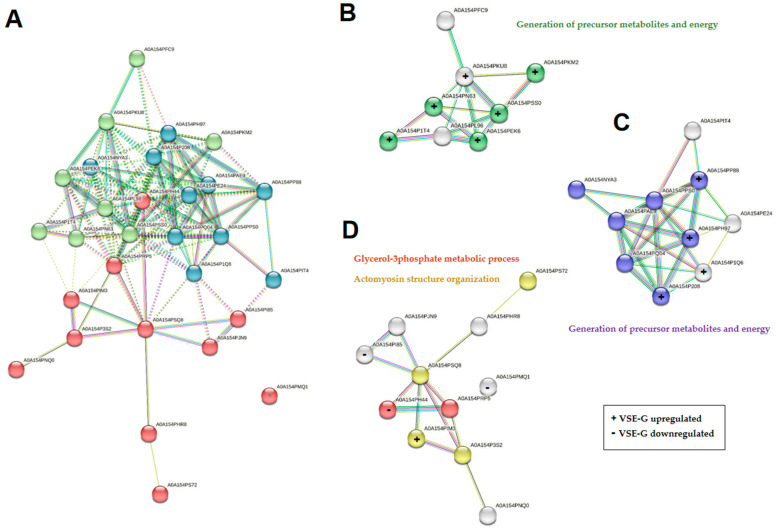
STRING protein–protein interaction (PPI) network analyses of the 29 proteins quantified from *Dufourea novaeanliae* in the venom sac extract (VSE) of the *Vespa crabro* gynes and workers. (**A**) Network clustered using k-means clustering: cluster 1 (green bubbles) with 8 proteins; cluster 2 (blue bubbles) with 10 proteins; and cluster 3 (red bubbles) with 11 proteins. The main biological process of proteins in cluster 1 (**B**), cluster 2 (**C**), and cluster 3 (**D**). The colored proteins within the clusters actively participate in the specified biological process. Protein classification relied on gene ontology (GO) biological processes. All referenced biological processes maintain a false discovery rate (FDR) of less than 0.05. VSE-G: venom sac extracts of gynes; VSE-W: venom sac extracts of workers. The nomenclature of proteins is based on software identification.

**Figure 7 ijms-25-03798-f007:**
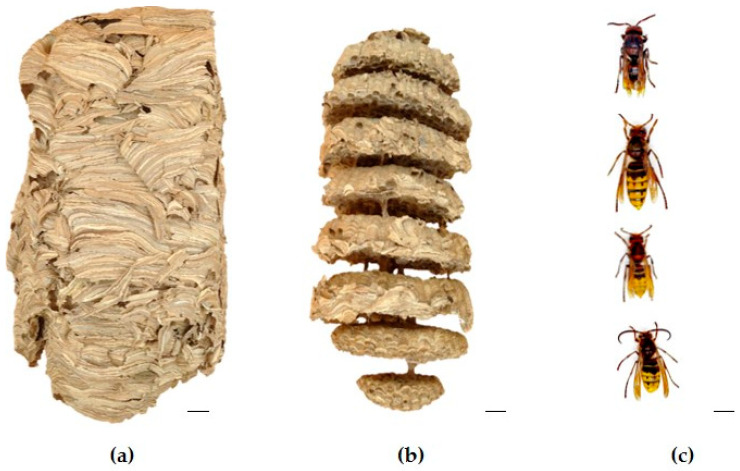
The sampled *Vespa crabro* nest after being removed: (**a**) lateral view; (**b**) lateral view after peeling away the paper envelope showing the internal chamber with the combs (*n* = 8). (**c**) From top to bottom: queen, gyne, worker, and male. Scales bars: a = 4 cm; b = 3 cm; c = 0.5 cm. Photo author: X. Feás.

**Table 1 ijms-25-03798-t001:** Classification of the 240 proteins (pp) found in the *Vespa crabro* venom sac extracts by SWATH-MS according to the species, family, order, and class to which they belong.

Class	Order	Family	Scientific Name	Common Name	nº	Group
*Actinopterygii*	*Cyprinodontiformes*	*Nothobranchiidae*	*Nothobranchius furzeri*	African Turquoise killifish	1	NOTFU
	*Perciformes*	*Siganidae*	*Siganus canaliculatus*	White-spotted spinefoot	1	SIGCA
*Amphibia*	*Anura*	*Dendrobatidae*	*Ranitomeya imitator*	Mimic poison frog	17	9NEOB
*Arachnida*	*Mesostigmata*	*Laelapidae*	*Tropilaelaps mercedesae*	Tropilaelaps mite	2	9ACAR
	*Scorpiones*	*Buthidae*	*Centruroides hentzi*	Hentz striped scorpion	2	9SCOR
		*Caraboctonidae*	*Hadrurus spadix*	Black back scorpion	1	
*Bacilli*	*Lactobacillales*	*Lactobacillaceae*	*Lactobacillus apis*	Lactic acid bacteria	1	9LACO
		*Streptococcaceae*	*Lactococcus lactis*	Lactic acid bacteria	1	9LACT
*Clitellata*	*Rhynchobdellida*	*Glossiphoniidae*	*Helobdella robusta*	Californian leech	1	HELRO
*Insecta*	*Coleoptera*	*Ptilodactylidae*	*Anchycteis velutina*	Toe-winged beetle	1	
	*Hymenoptera*	*Apidae*	*Apis cerana*	Asian honey bee	31	APICE
			*Apis cerana* subsp. cerana	Asian honey bee	4	APICC
			*Apis cerana* subsp. *indica*	Indian honey bee	1	
			*Apis mellifera*	European honey bee	41	APIME
			*Apis mellifera ligustica*	Italian honey bee	2	APILI
			*Bombus festivus*	Bumblebee of Sichuan	1	
			*Bombus humilis*	Brown-banded carder bumblebee	1	BOMHU
			*Bombus terrestris*	Buff-tailed bumblebee	44	BOMTE
			*Frieseomelitta varia*	Yellow marmalade bee	28	9HYME
		*Halictidae*	*Agapostemon leunculus*	Sweat bee	1	9COLE
			*Dufourea novaeangliae*	Sweat bee	29	DUFNO
		*Vespidae*	*Vespa basalis*	Black-bellied hornet	1	VESBA
			*Vespa crabro*	European hornet	1	VESCR
			*Vespa magnifica*	Magnifica hornet	1	VESMG
			*Vespa mandarinia*	Northern giant hornet	1	
			*Vespa mandarinia* subsp. *Japonica*	Northern giant hornet	1	VESMA
			*Vespa simillima* subsp. *xanthoptera*	Yellow hornet	4	VESXA
			*Vespa velutina*	Yellow-legged Asian hornet	6	VESVE
			*Vespula vulgaris*	Common wasp	1	VESVU
*Mammalia*	*Primates*	*Hominidae*	*Homo sapiens*	Human	2	HUMAN
*Microsporidia*	*Nosematida*	*Nosematidae*	*Nosema apis*	Nosema	1	9MICR
*Pisoniviricetes*	*Picornavirales*	*Iflaviridae*	Deformed wing virus	Deformed wing virus	1	9VIRU
*Reptilia*	*Squamata*	*Viperidae*	*Crotalus adamanteus*	Eastern diamondback rattlesnake	1	CROAD
*γ* *-proteobacteria*	*Enterobacterales*	*Enterobacteriaceae*	*Klebsiella* sp. *RIT-PI-d*	*Klebsiella*	1	9ENTR
			*Cronobacter sakazakii*	*Cronobacter*	1	CROSK
			*Citrobacter freundii*	*Citrobacter*	1	CITFR
			*Escherichia coli*	*coli*	5	ECOLX

**Table 2 ijms-25-03798-t002:** Proteins from other classes with dysregulated expression in the *Vespa crabro* gyne and worker venom sac extracts.

Family/Protein	Uniprot Code	Specie	*p*-Value (*t* Test)	FC (VSE-G/VSE-W)
Aldo-keto reductase/alcohol dehydrogenase (NADP(+))	J3SBS4	*adamanteus*	<0.001	16.364
Aldehyde dehydrogenase/Multifunctional fusion protein	A0A1W7RAT0	*H*. *spadix*	0.007	4.969
Heat shock protein/Heat shock 70 kDa protein	A0A2I9LP41	*C* *. hentzi*	0.004	2.081
Intermediate filament/hypothetical protein	A0A822AXZ5	*R*. *imitator*	0.027	0.458
Transferrin/Serotransferrin	A0A822GLX5	*R*. *imitator*	0.011	0.289
Apolipoprotein A1/A4/E/hypothetical protein	A0A822ID78	*R*. *imitator*	0.004	0.216
	A0A821UX93	*imitator*	0.006	0.465
Peptidase/Proteasome subunit alpha type	A0A821TAG2	*R*. *imitator*	<0.001	0.313
Globin/hypothetical protein	A0A822FRT2	*R*. *imitator*	0.004	0.265
Glyceraldehyde-3-phosphate dehydrogenase	A0A0A5SYN4	*C*. *freundii*	0.002	0.174
	A0A821LTU2	*R*. *imitator*	0.011	0.231
Lpp/Major outer membrane lipoprotein Lpp	A0A0L0AQN9	*Klebsiella* sp.	0.048	0.243
Thioredoxin/Thioredoxin	W8TGE9	*E*. *coli*	<0.001	0.214
No family described/Enolase	K4WKC0	*E*. *coli*	0.005	0.247
No family described/TetR family transcriptional regulator	A0A1C4ADY6	*L*. *apis*	<0.001	0.180
No family described/IgG H chain	S6B291	*H*. *sapiens*	<0.001	0.247
No family described/Ig-like domain-containing protein	Q8TCD0	*H*. *sapiens*	0.007	0.353
No family described/hypothetical protein	A0A822HYL0	*R*. *imitator*	0.005	0.571

VSE-G, venom sac extract of gynes; VSE-W, venom sac extracts of workers; FC (VSE-G/VSE-W), fold change of gyne/worker; FC < 1, downregulated protein in VSE-G; FC > 1, upregulated protein in VSE-G; statistical differences by *t* test, *p* < 0.05. The nomenclature of proteins is based on software identification.

## Data Availability

The mass spectrometry proteomics data have been deposited to the ProteomeXchange Consortium via the PRIDE partner repository with the dataset identifier PXD047955.

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
