# Peer review of "Peeking into the Stingers: A Comprehensive SWATH-MS Study of the European Hornet Vespa crabro (Linnaeus, 1758) (Hymenoptera: Vespidae) Venom Sac Extracts"

_ijms, 2024, doi:10.3390/ijms25073798_

Round 1
Reviewer 1 Report
Comments and Suggestions for Authors
The paper is really interesting, I suggest to red it better to avoid minor English error and some punctuation typos, Moreover the first time the authors wrote a species in the text, write it entirely (I suggest not to consider the specimen quoted in the table)
INTRODUCTION
Please add references to this sentence “This in-depth understanding of the venom proteome allows the identification, characterization, and functional annotation of venom proteins, enabling a holistic understanding of the venom's biological activities and potential applications in various fields” (es. https://doi.org/10.1038/s41598-021-84385-5; https://doi.org/10.1016/j.ibmb.2016.07.001)
Please add references to these sentences “EHs are social insects that live in large colonies. They build their nests in tree hollows, cavities in buildings, or other protected areas. The nest is made of a paper-like material created with wood fibers which are chewed up and mixed with saliva. As predators, EHs feed on their brood with a variety of insects, including other bees and wasps, flies, and caterpillars. Adults are also attracted to sweet substances and may be seen feeding on ripe fruit or 60 sugary liquids. While EH can be beneficial for controlling other insect populations, they can also be a nuisance or a danger to humans -and other animals.”
FIGURE
Figure 5, it could be useful to have an additional graph on the Cellular component category
Figure 8 is unnecessary
Figures 9 and 10 can be inserted as supplementary
MATERIAL AND METHODS
Lines 370, 380 please insert Quercus robur in italics
Please add a reference for the Bradford method
Comments on the Quality of English LanguageI suggest to red the paper better to avoid minor English error
Reviewer 2 Report
Comments and Suggestions for Authors
I have been revising the Manuscript "Peeking into the Stingers: A Comprehensive SWATH-MS 2 Study of the European Hornet Vespa crabro (Linnaeus, 1758) 3 (Hymenoptera: Vespidae) Venom Sac Extracts" by Feas et al.
The study analyses and compares the proteinaceous composition of the venom extract of Vespa crabro individuals pertaining to different casts by SWATH-MS 2.
The topic of the study is interesting and overall the MS conveys well the data obtained, however I have few concerns about the writing and editing of the MS.
1)It would probably be more appropriate to move subparagraph 2.1 and lines 111-113 of subparagraph 2.2 to the MM section
2)Figure 5: quality of image is low and it is not possible to read the text in the diagram. Please upload a higher quality image
3) When names of techniques are commonly shared you can use the acronym i.e. It is not necessary to write “Sodium dodecyl sulfate poly-134 acrylamide gel electrophoresis” in full, it is enough to write SDS-PAGE as it is commonly recognized in the scientific community
4)the MM section has too many subparagraphs some of which are really short, I advise to merge some of them to increase readability
5)Please, check out and correct scientific names of plants and animals, write them in italics and avoid repeating the full name after the first time i.e V. crabro. Also, once you have defined that European hornet and Vespa crabro are synonyms you can use just one, there is no need to repeat "European Hornet Vespa crabro (Linnaeus, 1758) every time. Please, go through the whole MS.
6) If original blots are going to be showed it would be useful to indicate the name of the samples and values in the ladder.
7)The Discussion section is quite confusing in writing and some of the ideas presented appear as long shots, please read thoroughly the section and improve readability of the text and tone down speculations.
Moreover, I have minor elements to point out:
Line 53: “The Vespa crabro (Linnaeus, 1758) (Hymenoptera: Vespidae), is a hornet, commonly known as the European hornet (EH). It is one of the largest species of hornet, measuring up to 35 mm in length, and is found in Europe, Asia and North America”.
Change in “Vespa crabro (Linnaeus, 1758) (Hymenoptera: Vespidae), commonly known as the European hornet (EH), is one of the largest species of hornet, measuring up to 35 mm in length, and is found in Europe, Asia and North America”
Line 58: I guess you wanted to say that they “Feed their brood with…”
Line 181: “allowing for an assessment of the biological processes” how was this assessed?
Line 213: D. novaeangliae does not pertain to the Helictidae family
Line 370: Quercus robur. Check how species of plants are written throughout the MS and adjust them when needed
Line 371: change as follows: sylvatica with
Line 386: “Eucalyptus logging points were identified within the study area and strategically selected as potential hotspots for vespid activity”. Is there a particular reason why you chose these as hotspots?
Line 394: “Easily distinguished from V. velutina by coloration in particular, the coloration of the second abdomen tergum (which has the posterior half almost entirely yellow, with the part clearly tridentate apically), and that of the fifth and sixth tergs (which are almost entirely yellow coloring.” Please rephrase the sentence as it is not very clear and beware of writing mistakes, i.e there is no verb in the sentence.
Line 398: how where individuals followed? Did you use any telemetry or did you follow them by walking?
Lines 403-404: Probably it is not necessary to include this information.
Line 425: It would be useful to have the total number of animals analyzed and then explain the process of creating pools.
Comments on the Quality of English Language
Minor editing of English language is required, particularly make sure that there is concordance between subject and verbs, and avoid sentences which are too long.
